# Skin Bacteria Mediate Glycerol Fermentation to Produce Electricity and Resist UV-B

**DOI:** 10.3390/microorganisms8071092

**Published:** 2020-07-21

**Authors:** Arun Balasubramaniam, Prakoso Adi, Tra My Do Thi, Jen-Ho Yang, Asy Syifa Labibah, Chun-Ming Huang

**Affiliations:** Department of Biomedical Sciences and Engineering, National Central University, Taoyuan 32001, Taiwan; arunb2895@gmail.com (A.B.); clever.black99@gmail.com (P.A.); tramydo0510@gmail.com (T.M.D.T.); howard10000237@gmail.com (J.-H.Y.); syifafafisika13@gmail.com (A.S.L.)

**Keywords:** electricity, fermentation, ferrozine, *Staphylococcus epidermidis* (*S. epidermidis*), ultraviolet-B (UV-B)

## Abstract

Bacteria that use electron transport proteins in the membrane to produce electricity in the gut microbiome have been identified recently. However, the identification of electrogenic bacteria in the skin microbiome is almost completely unexplored. Using a ferric iron-based ferrozine assay, we have identified the skin *Staphylococcus epidermidis* (*S. epidermidis*) as an electrogenic bacterial strain. Glycerol fermentation was essential for the electricity production of *S. epidermidis* since the inhibition of fermentation by 5-methyl furfural (5-MF) significantly diminished the bacterial electricity measured by voltage changes in a microbial fuel cell (MFC). A small-scale chamber with both anode and cathode was fabricated in order to study the effect of ultraviolet-B (UV-B) on electricity production and bacterial resistance to UV-B. Although UV-B lowered bacterial electricity, a prolonged incubation of *S. epidermidis* in the presence of glycerol promoted fermentation and elicited higher electricity to suppress the effect of UV-B. Furthermore, the addition of glycerol into *S. epidermidis* enhanced bacterial resistance to UV-B. Electricity produced by human skin commensal bacteria may be used as a dynamic biomarker to reflect the UV radiation in real-time.

## 1. Introduction

The human skin, the largest organ in the body, is annexed by a diverse range of microorganisms that are competent in extracellular electron transfer (EET) during electricity production [1,2]. Bacteria, such as *Staphylococcus epidermidis* (*S. epidermidis*) and *Staphylococcus capitis* (*S. capitis*), in human skin have demonstrated distinct electrogenic capabilities which are directly analogous to those of the well-known Gram-negative exoelectrogens [2,3]. Bacteria are generally perceived to yield electrons intracellularly and transport these electrons to their acceptors in the extracellular region. Electrons are usually yielded in the bacteria by oxidation of electron donors, such as acetate, and transfer them to other microorganisms, or to electron acceptors such as ferric iron (Fe^3+^) or manganese ion 3^+^ or 4^+^ (Mn^3+^ or Mn^4+^) [4,5]. Data from extensive studies have demonstrated that electrogenic Gram-positive bacteria, such as *Listeria monocytogenes* (*L. monocytogenes*) and *Enterococcus faecalis* (*E. faecalis*), and many probiotic *Lactobacillus* species in the human gut expressed genes encoding proteins responsible for the EET process that enhanced bacterial growth [6,7,8]. The gene-encoding proteins for the EET family of homologs are believed to present in both Gram-negative and Gram-positive bacteria [7]. The EET properties of bacteria can be determined by electrochemical methods. Microbial fuel cell (MFC) technology is one of the bio-electrochemical techniques that have great potential to study the electrogenic properties of bacteria [9,10]. Microorganisms in the MFCs possess complete enzymatic pathways as part of their natural metabolism to regenerate biocatalytic enzymes. As a consequence, they have operating resources that can be self-assembled, self-repaired, and self-maintained [2]. The magnitude of electricity production in MFCs depends on some factors such as the selection of electrogenic bacteria, electron donors, and electron acceptors. Generally, pure or mixed bacteria were used for the study of electricity production in MFCs [11]. Although MFC can be used as a device to determine the electrogenicity of bacteria, it is not a high-throughput and rapid screening tool for the selection of electrogenic bacteria from the human microbiome.

Bacteria in the skin microbiome can mediate fermentation to preserve skin homeostasis that influences the skin immunity in a similar way to bacteria in the gut microbiome [12]. Recent studies have found that skin microbiome constricted ultraviolet (UV)-induced immune intimidation. Specifically, it has been reported that 6-N-hydroxyaminopurine, an *S. epidermidis* metabolite, can deliberately hamper UV-induced skin tumors in mice [13]. In the current study, we establish a ferrozine assay [13] in agar plates which allows us to rapidly screen the electrogenic bacteria in both fecal and skin microbiomes. Electrogenic *S. epidermidis* in the skin microbiome was selected to further characterize its electricity production via glycerol fermentation. We found that, although UV-B affected the electricity production of *S. epidermidis*, bacteria can mediate glycerol fermentation to provoke higher electricity to resist UV-B. 

## 2. Materials and Methods 

### 2.1. Ethics Statement

This study was performed in accordance with the protocols (NCU-106-016, 19 December 2017) of the Institutional Animal Care and Use Committee (IACUC) of the National Central University (NCU). The Institutional Review Board (IRB) (No. 19-013-B1, 22 May 2019) at Landseed International Hospital, Taiwan, approved the consent procedure for fecal and skin swab samplings. Both fecal and skin swab samples were collected from three healthy subjects. Written consent was obtained from all participants prior to inclusion in the study.

### 2.2. Bacteria Identification

The fecal and skin swab samples were streaked by plating on a tryptic soy broth (TSB) (Sigma, St. Louis, MO, USA) agar plate supplemented with 2% glycerol or glucose, and 0.1 mg/mL ferric ammonium citrate (FAC) at 37 °C for 24 h. Plates were then taken out of the incubator and an overlay assay of 10-ml ferrozine media (0.8% agarose and 2 mM ferrozine) was applied. The color change of colonies in agar plates from light (ferric ammonium citrate) to dark purple (ferrozine-chelatable irons) indicated the electron production. The colonies were selected and cultured in 10 ml of TSB media for 24 h at 37 °C. The genomic DNA was extracted from the selected colonies using an Easy Pure Genomic DNA Spin Kit (Bioman, New Taipei, Taiwan). Sequence analysis of 16S ribosomal RNA (rRNA) genes was utilized for bacterial identification. The identification of bacterial strains was validated by rRNA sequencing using the 16S rRNA 27F and 534R primers for a polymerase chain reaction (PCR). The sequencing result was further analyzed by the Basic Local Alignment Search Tool (BLAST) for species identification. An identified *S. epidermidis* (10^7^ colony-forming unit (CFU)) or *S. epidermidis* ATCC 12228 (10^7^ CFU) was spotted on TSB agar plates for ferrozine assay. The color change in bacterial colonies was observed. 

### 2.3. Microbial Fuel Cells (MFC)

The MFC was constructed with acrylic sheets of 8 cm × 8 cm × 5.5 cm with a total volume of 352 mL. The cathode and proton exchange membrane (PEM) (Nafion N117, Homy Tech, Taipei, Taiwan) were crafted by press forming, in which pieces of carbon cloth (Homy Tech) that were 5 × 5 cm in length and width. The anode was fabricated by a carbon felt (Homy Tech) of 5 × 5 cm in length and width (10 mm thickness). A copper wire was applied to connect the anode and cathode (0.4 mm thickness). An external resistance (1000 Ω) was set between the anode and cathode to facilitate electrode transmission. Rich media (250 mL) supplemented with 10^7^ CFU/mL *S. epidermidis* ATCC 12228 in the presence or absence of 2% glycerol were incubated in MFC at 37 °C for 24 h. A digital multimeter (Lutron, DM-9962SD, Sydney, Australia) was utilized to periodically measure the voltage changes using a data logger.

### 2.4. Bacterial Fermentation

*S. epidermidis* ATCC 12228 (10^7^ CFU/mL) were incubated in 10 mL of rich media (10 g/L yeast extract (Biokar Diagnostics, Beauvais, France), 3 g/L TSB, 2.5 g/L K_2_HPO_4_, and 1.5 g/L KH_2_PO_4_) in the absence and presence of 2% glycerol at 37 °C for one or three day(s). Rich media with 2% glycerol alone or bacteria alone were included as controls. The 0.002% (*w*/*v*) phenol red (Sigma) in rich media has proceeded as a fermentation indicator. A color change from red-orange to yellow determined the phenomenon of bacterial fermentation. The color change was quantified by absorbance at 562 nm. In some experiments, *S. epidermidis* bacteria were pre-incubated with 0.04% 5-methyl furfural (5-MF), a furfural analog as a fermentation inhibitor, at 37 °C for 12 h before experiments were carried out.

### 2.5. Electricity Detection 

Electricity produced by bacteria was detected by using a chamber equipped with cathode and anode (Appendix A). A carbon felt (2.5 × 10 cm) and a carbon cloth (10 × 10 cm) (Homy Tech) were respectively used for fabricating anode and cathode. The cathode has been wrapped up to a Nafion membrane N117 (6 × 6 cm) (Homy Tech) which was served as a PEM. Copper wires were used to connect anode and cathode with an external resistance (1000 Ω). Bacteria or their respective controls were pipetted on the surface of the anode. Electricity was recorded by the changes in voltage (mV) against time (min) using a digital multimeter. The recorded voltages in every 10 s were used for plotting a graph. 

### 2.6. UV-B Irradiation 

*S. epidermidis* ATCC 12228 (10^7^ CFU/mL) were irradiated at 0, 0.5, 1, 5, and 10 mJ/cm^2^ dosage using a UV-B lamp (Model EB-280C, Spectronics Corp., Westbury, NY, USA). The irradiated bacteria were then incubated in a 1.5 ml Eppendorf tube at 37 °C for 12 h to count CFUs. The bacteria sample was serially diluted 1: 10^0^–1:10^5^ in a 96 well plate and 10 μl of serially diluted bacteria were dropped onto TSB agar plates. 

### 2.7. Statistical Analysis

GraphPad Prism® software was employed for data analysis by unpaired *t*-test. The significant difference was considered by *p*-values observation as follows: *p*-values of <0.05 (*), <0.01 (**), and <0.001 (***). At least three separate experiments were used for displaying the mean ± standard deviation (SD).

## 3. Results

### 3.1. A TSB Agar-Based Ferrozine Overlay Assay for Screening Electrogenic Bacteria

When the exposure of FAC to the electron produced by bacteria caused the reduction of Fe^3+^ to Fe^2+^, that bound to ferrozine and formed dark purple complexes abilities [7,14]. We mixed FAC (Fe^3+^) with TSB media, agarose, and ferrozine to create solid agar plates. To screen the electrogenic bacteria, the plates supplemented with 2% glucose or glycerol, respectively, were overlaid with samples collected from human fecal and skin. Previous studies have verified glucose and glycerol as electron donors for electricity production [15]. As shown in Figure 1a,b, several bacterial colonies formed the dark purple (ferrozine-Fe^2+^) complexes after 24 h incubation. Three dark purple colonies and one light purple colony from both fecal and skin samples were randomly selected and labeled as F1-4 and S1-4, respectively. F1, F2, and F3 colonies with dark purple complexes in fecal samples were identified as *E. faecalis* by 16S rRNA sequencing (Appendix A). The 16S rRNA sequence of the F4 colony with light purple color matched with that of *Enterococcus avium* (*E. avium*). The 16S rRNA of S1 and S2 colonies with dark purple complexes in skin samples were sequenced as *S. epidermidis*. The S3 colony with a dark purple complex was verified as *Staphylococcus hominis* (*S. hominis*) and the S4 colony with light purple color was confirmed as *Pantoea vagans* (*P. vagans*) (Appendix A).

To validate the activity of *S. epidermidis* at the ferric iron (Fe^3+^) reduction, *S. epidermidis* ATCC 12228 and *S. epidermidis* selected from the S1 colony (10^7^ CFU) were spotted on TSB agar-based ferrozine plates supplemented with 2% glycerol for 24 h. The dark purple complexes in colonies of *S. epidermidis* ATCC 12228 and *S. epidermidis* S1 isolate indicated that *S. epidermidis* is an electron-producing bacterium which can convert Fe^3+^ to Fe^2+^ (Figure 1c). Our previous studies have shown that *S. epidermidis* can ferment glycerol and produce short-chain fatty acids such as acetate and butyrate [16] which have been documented as potent electron donors [17]. To examine whether the activity of ferric iron (Fe^3+^) reduction by *S. epidermidis* is mediated by fermentation, bacteria were pretreated with 5-MF, a furfural analog as a fermentation inhibitor, which can inhibit acetolactate synthase (ALS) and other enzymes in the fermentation pathway of bacteria [12,18]. The colony of 5-MF-pretreated *S. epidermidis* appeared light purple, indicating that the activity of ferric iron (Fe^3+^) reduction by *S. epidermidis* was mediated by glycerol fermentation (Figure 1c). *E. faecalis,* one of the bacteria in the gut microbiome, have been studied extensively for electron production via EET [19,20,21,22]. The activity of ferric iron (Fe^3+^) reduction by *S. epidermidis* displayed the potential of skin *S. epidermidis* as an electrogenic bacterium. Data in Figure 1 also highlight the value of using TSB agar-based ferrozine overlay assays for screening electrogenic bacteria in the human microbiome. 

### 3.2. Electricity Production of S. epidermidis in MFCs

To further validate the electrogenic property of *S. epidermidis*, the electricity measured by voltage change (mV) in the incubation of bacteria in rich media in MFCs were monitored for 70 min (Figure 1a). The media alone did not alter the changes in voltage. *S. epidermidis* (10^7^ CFU/mL) in rich media (250 mL) which contains 0.25% dextrose in TSB generated a voltage change with a peak at approximately 100 mV. The addition of 2% glycerol into *S. epidermidis* incubation resulted in a two-fold increase in voltage changes (Figure 2b). The data demonstrated that glycerol can enhance the electricity production of *S. epidermidis*. No voltage, or almost none, was detected when 5-MF-pretreated *S. epidermidis* in the presence of 2% glycerol was added into MFCs, demonstrating that electricity production of *S. epidermidis* was mediated by glycerol fermentation. 

### 3.3. The Effect of UV-B on Electricity Production of S. epidermidis 

We first examined the glycerol fermentation *of S. epidermidis*. *S. epidermidis* (10^7^ CFU/mL) was cultured in rich media containing phenol red in the presence of 2% glycerol for 24 h. Media with glycerol alone or bacteria alone acted as controls. A change in color of phenol red from red to yellow and a significant reduction of the optical density of 562nm (OD_562_) due to low pH values [23] in the culture media of the *S. epidermidis* in the presence of 2% glycerol were detected, indicating the occurrence of bacterial fermentation (Figure 3a). The OD_562_ value in a 72 h-culture of *S. epidermidis* in the presence of 2% glycerol was considerably lower than that in a 24 h-culture, suggesting that the prolonged culture of *S. epidermidis* with glycerol may increase the production of SCFAs in fermentation media. 

Human skin and its microbial inhabitants can be simultaneously exposed to environmental stressors, such as UV radiation. It has been reported that UV-B exposure can influence the growth of skin bacteria and change the homeostasis of the skin microbiome [24]. UV-B exposures can damage the electron transport mediated by Photosystem II in plants [25] and interrupt the electron flow in the mitochondrial membrane in mammalian cells [26]. Here, we investigated if the electricity production of *S. epidermidis* can be influenced by UV-B exposure. Unlike MFCs with a 250 ml tank, we fabricated a small-scale (5 mL) *in vitro* chamber (Appendix A) equipped with both an anode and cathode. Since the *S. epidermidis* was directly applied onto the surface of the anode during electricity detection, allowing bacteria to be fully exposed to UV-B. *S. epidermidis* was incubated in rich media in the absence or presence of 2% glycerol for 24 h before application onto anode for detection of voltage changes for 20 min. As shown in Figure 3b, the application of *S. epidermidis* with or without glycerol onto the anode yielded a voltage change with a peak at 2.6 and 1.7 mV, respectively. When bacteria were exposed to 1 mJ/cm^2^ UV-B during detection, the voltages were significantly reduced, indicating that UV-B can impede the electricity production of *S. epidermidis*. Since fermentation became pronounced when the incubation time of *S. epidermidis* in the presence of glycerol was extended from 24 to 72 h (Figure 3a), we examined the electricity production and UV-B sensitivity of *S. epidermidis* after 72 h incubation. The surface of the anode was applied with *S. epidermidis* with or without glycerol after 72 h incubation. As shown in Figure 3b, the voltage induced by *S. epidermidis* with or without glycerol for 72 h incubation was significantly higher than that induced by bacteria with or without glycerol for 24 h incubation. Furthermore, during voltage detection, the exposure of *S. epidermidis* from 72 h incubation to 1 mJ/cm^2^ UV-B did not attenuate the magnitude of the voltage induced by bacteria. The data suggested that prolonged incubation may increase the amounts of SCFAs in fermentation media. As electron donors, SCFAs in fermentation media may enhance the capability of *S. epidermidis* to produce higher and long-duration electricity which prevents electricity loss caused by UV-B exposure. 

### 3.4. Enhancement of Bacterial Resistance to UV-B by Glycerol Fermentation 

Glycerol, a trihydroxy alcohol, has been used in topical dermatological formulations to protect skin from UV irradiation [27]. Data in our previous studies revealed that glycerol served as a carbon source to induce fermentation of bacteria in the skin microbiome [28]. We here examined if glycerol fermentation of *S. epidermidis* alters the bacterial sensitivity to UV-B. *S. epidermidis* (10^7^ CFU/mL) with and without 2% glycerol was irradiated by UV-B at 0, 0.5, 1, 5, and 10 mJ/cm^2^ for 12 h. As shown in Figure 4a, more than 1 log_10_ reduction at bacterial number (0 mJ/cm^2^; 2.3 ± 0.6 10^7^ CFU/mL versus 5 mJ/cm^2^; 3.0 ± 0.1 10^6^ CFU/mL) was detected when *S. epidermidis* was exposed to 5 mJ/cm^2^ UV-B. *S. epidermidis* was completely killed by 10 mJ/cm^2^. The addition of 2% glycerol into *S. epidermidis* culture during UV-B exposure significantly prevented the reduction of bacterial number by 5 mJ/cm^2^ UV-B. However, the prevention by the addition of glycerol became ineffective when *S. epidermidis* was pretreated with 5-MF (Figure 4b). These data demonstrated that the glycerol fermentation of *S. epidermidis* enhanced bacterial resistance to UV-B. 

## 4. Discussion

Although it has been reported that an electrode-based mechanism called electro-fermentation (EF) can control microbial fermentation [29], we demonstrated for the first time that bacteria can mediate fermentation to generate electricity. A TSB agar-based ferrozine overlay assay was employed to identify the electrogenic bacteria in the human microbiome. Three Gram-positive bacteria (*E. faecalis*, *S. epidermidis,* and *S. hominis*) developing the dark purple (ferrozine-Fe^2+^) complexes exhibited their activities of ferric iron (Fe^3+^) reduction (Figure 1). In agreement with previous studies [30], we identified *E. faecalis* as an electrogenic bacterium which can elicit more than 8 mV voltage in the presence of 2% glucose (Appendix A). Here, we demonstrated that two skin bacteria (*S. epidermidis* and *S. hominis*) can provoke approximately 3 mV voltage in the presence of 2% glycerol within 20 min in a small-scale in vitro chamber (Figure 3b and Appendix A). Unlike Gram-negative bacteria, most Gram-positive bacteria with a thick peptidoglycan layer cannot synthesize heme. However, in *E. faecalis*, two canonical heme proteins, catalase [31] and cytochrome *bd* [32,33], can be assembled once bacteria were supplied with heme from the environment. The cytochrome oxidizes the reduced demethylmenaquinone (DMK) to reduce molecular oxygen to water. A type II NADH: quinone oxidoreductase and various membrane-associated dehydrogenases play a role in the reduction of DMK [34,35]. Although *E. avium* in the fecal sample and *P. vagans* in the skin swab sample formed light purple complexes in the presence of glucose or glycerol (Figure 1), we cannot rule out the possibility that those bacteria can mediate other carbon sources for electricity production. Several bacteria such as *Bacillus subtilis*, *Flavobacterium* sp., *Aeromonas hydrophila*, *Citrobacter freundii*, and *Stenotrophomonas *sp., have been reported as electrogenic microorganisms [36]. However, most bacteria identified from the TSB agar-based ferrozine overlay assay were *Staphylococcus* and *Enterococcus* species. One of possible reasons was that some bacteria with sensitivities to irons may be unable to grow on agar plates which contained ferric ammonium citrate.

The mechanism of electricity produced by *S. epidermidis* is unclear. Our results demonstrated that fermentation is essential for the electricity production of *S. epidermidis* since inhibition of the fermentation process by 5-MF remarkably reduced the voltage elevation (Figure 2). Furfural, a fermentation product, and its analog 5-hydroxy methyl furfural (5-HMF) can inhibit glycolytic and fermentative enzymes such as alcohol dehydrogenase (ADH) and aldehyde dehydrogenase (ALDH) and pyruvate dehydrogenase (PDH) [37]. Results in our publication demonstrated that 5-MF can effectively inhibit ALS in *S. epidermidis* [37] SCFAs such as acetic acid and butyric acid was detectable in the media of glycerol fermentation of *S. epidermidis* [16]. The inhibition of glycerol fermentation of *S. epidermidis* by 5-MF may lead to a decrease in the production of SCFAs. Gram-positive bacteria, in general, are poor in current production. However, in the presence of electron donors, such as glycerol, acetic acid, butyric acid, or ethanol, these bacteria can mediate a flavin-based EET [7] system to donate electrons to external electron acceptors [38]. Our future work will determine if *S. epidermidis* uses the flavin-based EET system to transport electrons induced by SCFAs. 

UV-B radiation represents the most cytotoxic waveband of solar radiation that can induce reactive oxygen species (ROS) to damage the bacteria [39]. Although the mechanism of bacterial resistance to UV-B is not fully explored, bacterial heterogeneity in susceptibility to UV-B radiation could be due to differences in their abilities to cope with oxidative damage. Gram-positive bacteria have been proposed to be more resistant to UV radiation than Gram-negative bacteria due to their cell wall characteristics, spore- or biofilm-forming capacity or low AT content, recognizing that thymine dimmers are the main target in UV-B damage [40]. The exposure of *S. epidermidis* ATCC 12228, a non-biofilm forming bacterium [20], with UV-B greater than 5 mJ/cm^2^ for 12 h significantly suppressed bacterial growth (Figure 4). However, the UV-B-induced suppression of *S. epidermidis* growth can be attenuated when bacteria were incubated with 2% glycerol. In the presence of glycerol, the culture of *S. epidermidis* in rich media for 24 h generated detectable electricity which can be enhanced when the culture was prolonged to 72 h (Figure 3). The function of electricity produced by *S. epidermidis* in the human skin microbiome is not yet defined. Although the concept of electrons as antioxidants was proposed [41], further validation of free radical neutralization by electrons produced by bacteria is required. Electrogenic bacteria may mediate fermentation to accumulate SCFAs as electron donors to enhance the electricity against a noxious environment [42,43]. Our results collected from Figure 3 and Figure 4 support this concept since the addition of glycerol into the culture of *S. epidermidis* promoted bacterial fermentation, electricity production, and resistance to UV-B. 

## 5. Conclusions

In the current study, we provided a novel method by using a TSB agar-based ferrozine overlay for screening the electrogenic bacteria in the human microbiome. We demonstrate for the first time that skin *S. epidermidis* bacteria can mediate glycerol fermentation to elicit electricity and enhance their resistance to UV-B. 

## Figures and Tables

**Figure 1 microorganisms-08-01092-f001:**
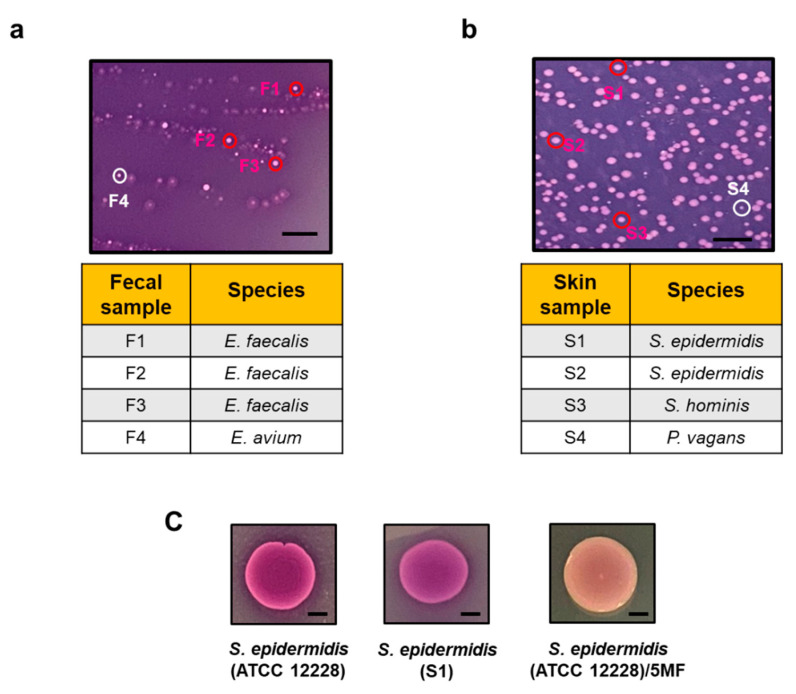
A TSB agar-based ferrozine overlay assay for screening electrogenic bacteria. (**a**) A representative image of the ferric iron (Fe^3+^) reduction by bacteria in human fecal samples. The colonies which turn their colors from light to dark purple were randomly marked as F1, F2, and F3. A colony with light was labeled as F4. To establish their species, all selected colonies were sequenced and listed. (**b**) Image of the ferric iron reduction by bacteria in skin swabs. The colonies with dark purples were marked as S1, S2, and S3. A colony with light purple was indicated as S4. 16S rRNA sequencing was conducted for the identification of bacterial species for each colony (Appendix A). (**c**) Images of ferric iron reduction of *S. epidermidis* strain ATCC 12228, S1, and 5-MF-pretreated ATCC 12228 on TSB agar-based ferrozine overlay assays. (scale bars = 5 mm).

**Figure 2 microorganisms-08-01092-f002:**
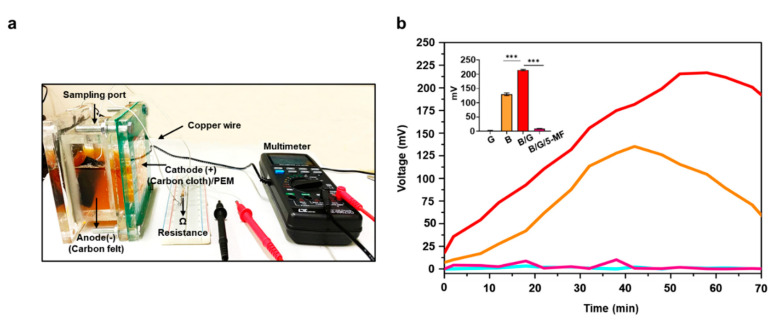
*Electricity Production of S. epidermidis in MFCs*. (**a**) A representative image of *S. epidermidis* ATCC 12228 plus 2% glycerol in MFC equipped with carbon felt as an anode, carbon cloth/PEM as a cathode, 1000 Ω resistance, sample port, copper wire, and a multimeter. (**b**) Electricity measured by voltage changes (mV) in MFC was recorded for 70 min in media containing glycerol (G), bacteria (B), bacteria plus glycerol (B/G), or 5-MF-pretreated bacteria plus glycerol (B/G/5-MF). The inset panel represented the peak voltage of each group. Data are the mean ± SD from three separate experiments. *** *p* < 0.001 (two-tailed *t*-tests).

**Figure 3 microorganisms-08-01092-f003:**
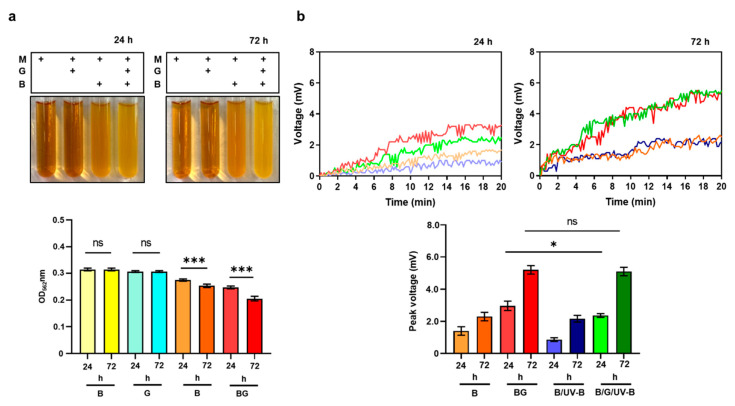
The effect of UV-B on electricity generation from a 24 or 72 h period of *S. epidermidis* fermentation. (**a**) Media (M) alone or with 10^7^ CFU/mL bacteria (B) in the presence or absence of 2% glycerol (G) were incubated for 24 or 72 h. The color change of phenol red in media from red to yellow indicated the phenomenon of fermentation which was quantified by OD_562_. (**b**) Electricity measured by voltage changes (mV) in an *in vitro* chamber was recorded for 20 min upon a UV-B dosage of 1 mJ/cm^2^. The peak voltage of each group was quantified. Results are the mean ± SD from three independent experiments. ns = non-significant. * *p* < 0.05; *** *p* < 0.001 (two-tailed *t*-tests).

**Figure 4 microorganisms-08-01092-f004:**
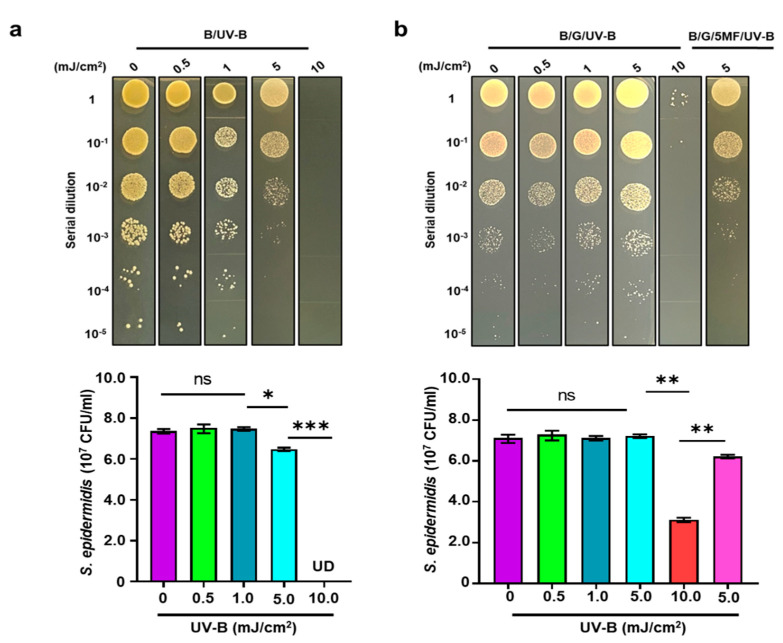
Enhancement of Bacterial Resistance to UV-B by Glycerol Fermentation. The photographic images of (**a**) 10^7^ CFU *S. epidermidis* (B) in the absence (**a**) or presence (**b**) of 2% glycerol (G) was irradiated by different UV-B dosages of 0, 0.5, 1, 5, and 10 mJ/cm^2^. 5-MF-pretreated *S. epidermidis* (10^7^ CFU) in the presence of 2% glycerol (5-MF/B/G/UV-B) was irradiated by 5 mJ/cm^2^. CFUs of bacteria with/without UV-B irradiation were counted after dropping the serially diluted (1:10^0^–1:10^5^) bacteria onto TSB agar plates and illustrated by the mean ± SD, in triplicate. ns = non-significant. * *p* < 0.05; ** *p* < 0.01 *** *p* < 0.001 (two-tailed *t*-tests).

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
