# Peer review of "Skin Bacteria Mediate Glycerol Fermentation to Produce Electricity and Resist UV-B"

_microorganisms, 2020, doi:10.3390/microorganisms8071092_

Round 1
Reviewer 1 Report
I would like to congratulate the authors to great and sound study. I enjoyed reading it and consider EET a very important scientific topic. Before publishing I would suggest a few minor changes.
Section 3.3. OD 562nm and Phenol red
Maybe clarify here a bit that the absorbance of phenol red at 562nm is representing the pH of the solution. It can be confusing for readers with different backgrounds to understand this section. For example a scientist who is used to work with mammalian cells will miss in the Figure 3 the strong red color from phenol red as it is seen in tissue cultures. A microbiologist unfamiliar with phenol red could mix up the OD 562nm with the commonly used OD 600nm which describes turbidity. The microbiologist might then wonder why a decrease in OD562 is representing fermentation. One way could be to speak about absorbance at 562nm instead of OD and then briefly in one sentence clarify that this absorbance represents change in pH before stating the results.
Line 203 it seems the word “glycerol” is missing after the 2%
In the discussion section the authors suggest "literature has recognized electrons as antioxidants to neutralize free radicals [41]." I find the cited publication weak. It is only a review in "THE JOURNAL OF ALTERNATIVE AND COMPLEMENTARY MEDICINE" which I am unfamiliar with. Also the main concept of grounding discussed in the review is puzzling to me and I found quite a bit of discussion on this topic. Primary literature on this was never published well. I would therefore suggest to the authors to either provide more citations on the point that free electrons can act as antioxidant, mark it more as a speculation or remove the statement.
In my eyes EET is by itself already an extremely exciting phenomena which reaches in my opinion beyond the simple use as fuel cells. It would not even be necessary to connect this to the antioxidant concept. That this could now be proven in skin commensal bacteria is very exciting and I am looking forward to the follow up work of the authors.
Author Response
Responses to reviewers
We appreciate the informative and insightful comments written by the reviewers on our manuscript (Microorganisms-876684) entitled, “Skin Bacteria Mediate Glycerol Fermentation to Produce Electricity and Resist UV-B “. Responses to reviewers point-by-point can be found as below. Changes have been underlined in the text of the manuscript.
Reviewer 1
Comment 1:
Comments and Suggestions for Authors
I would like to congratulate the authors to great and sound study. I enjoyed reading it and consider EET a very important scientific topic. Before publishing I would suggest a few minor changes.
Section 3.3. OD 562nm and Phenol red
Maybe clarify here a bit that the absorbance of phenol red at 562nm is representing the pH of the solution. It can be confusing for readers with different backgrounds to understand this section. For example, a scientist who is used to work with mammalian cells will miss in the Figure 3 the strong red color from phenol red as it is seen in tissue cultures. A microbiologist unfamiliar with phenol red could mix up the OD 562nm with the commonly used OD 600nm which describes turbidity. The microbiologist might then wonder why a decrease in OD562 is representing fermentation. One way could be to speak about absorbance at 562nm instead of OD and then briefly in one sentence clarify that this absorbance represents change in pH before stating the results.
Response 1:
Answer: Sentence has been fixed and highlighted with an additional reference cited.
Line 189,190 :
A change in color of phenol red from red to yellow and a significant reduction of the optical density of 562nm (OD562) due to low pH values [23] in the culture media of the S. epidermidis in the presence of 2% glycerol were detected, indicating the occurrence of bacterial fermentation (Figure 3a).
Reference 23: Kao, M.S.; Huang, S.; Chang, W.L.; Hsieh, M.F.; Huang, C.J.; Gallo, R.L.; Huang, C.M. Microbiome precision editing: Using PEG as a selective fermentation initiator against methicillin-resistant Staphylococcus aureus. Biotechnol J 2017, 12, doi:10.1002/biot.201600399.
Comment 2:
Line 203 it seems the word “glycerol” is missing after the 2%
Response 2:
Answer: The missing word “glycerol” has been added inline 203.
“absence or presence of 2% glycerol for 24 h before application onto anode for detection of voltage.”
Comment 3:
In the discussion section the authors suggest "literature has recognized electrons as antioxidants to neutralize free radicals [41]." I find the cited publication weak. It is only a review in "THE JOURNAL OF ALTERNATIVE AND COMPLEMENTARY MEDICINE" which I am unfamiliar with. Also, the main concept of grounding discussed in the review is puzzling to me and I found quite a bit of discussion on this topic. Primary literature on this was never published well. I would therefore suggest to the authors to either provide more citations on the point that free electrons can act as antioxidant, mark it more as a speculation or remove the statement.
In my eyes EET is by itself already an extremely exciting phenomena which reaches in my opinion beyond the simple use as fuel cells. It would not even be necessary to connect this to the antioxidant concept. That this could now be proven in skin commensal bacteria is very exciting and I am looking forward to the follow up work of the authors.
Response 3:
Answer: We totally agree reviewer’s comment. The sentence at the end of Discussion has been rewritten as below.
“The function of electricity produced by S. epidermidis in the human skin microbiome is not yet defined. Although the concept of electrons as antioxidants was proposed [43], further validation of free radical neutralization by electrons produced by bacteria is required.”
Reviewer 2
Comment 1:
Comments and Suggestions for Authors
The authors use a ferrozine assay to identify electrogenic bacteria from human fecal matter and skin swab samples. They identify Enterococcus faecalis, Staphylococcus epidermidis, and Staphylococcus hominis as electrogenic. While the first two bacteria have been previously identified as electrogenic, the main novelty of the manuscript is the implication of glycerol fermentation for electricity production. The authors show that the voltage in a microbial fuel cell increases when glycerol is added and decreases when the 5-methyl furfural is added. Interestingly, glycerol addition enhanced resistance of S. epidermidis to UV-B radiation as well.
This is certainly an interesting topic that merits exploration, but there are a few issues that should be addressed:
1) Regarding Figure 1a-b, the text refers to picking dark colonies to identify electrogenic bacteria. However, in Fig. 1a, it looks like they picked the lightest colonies. In Fig 1b, they all look light. This needs to be explained.
Response 1:
Answer: Identification of Enterococcus faecalis which has known as an electrogenic bacterium validates the feasibility of using TSB agar-based ferrozine overlays for screening the novel electrogenic bacteria in the human microbiome.
We randomly selected four colonies with dark purple complexes and one colony with a light purple complex as a control. Although bacteria formed light purple complexes may be poorly electrogenic in the presence of glucose or glycerol, we cannot rule out those bacteria can mediate other carbon sources for electricity production.
The description has been added onto the end of the first paragraph of Discussion Section.
“Although E. avium in the fecal sample and P. vagans in the skin swab sample formed light purple complexes in the presence of glucose or glycerol (Figure 1), we cannot rule out the possibility that those bacteria can mediate other carbon sources for electricity production.”
Comment 2:
2) There are minor typos and language issues in the paper, although it is otherwise well-written.
Response 2:
Answer: We have fixed minor typos and language issues in the revised manuscript.
Comment 3:
3) The ferrozine assay is nice in that many different bacteria can be screened. Why did the authors only pick 4 colonies from each sample? Given how few colonies were tested, I wonder why the authors even bothered with human sampling as opposed to using publicly available strains that are known to be electrogenic. Can the authors at least comment on the likelihood that other taxa besides Staphylococcus and Enterococcus are electrogenic?
Response 3:
Answer: The valuable comment has been added onto the end of first paragraph of Discussion Section. One new reference has been cited.
“Several bacteria such as Bacillus subtilis, Flavobacterium sp., Aeromonas hydrophila, Citrobacter freundii, and Stenotrophomonas sp., have been reported as electrogenic microorganisms [36]. However, most bacteria identified from the TSB agar-based ferrozine overlay assay were Staphylococcus and Enterococcus species. One of possible reasons was that some bacteria with sensitivities to irons may be unable to grow on agar plates which contained ferric ammonium citrate.
Editor
References 35 and 36 are the same.
Please revise, or please make a note to us to edit. Kindly note that we can help to remove the duplicated one and renumber all the references.
Response 1:
Answer: The duplicated references have been removed.

Reviewer 2 Report
The authors use a ferrozine assay to identify electrogenic bacteria from human fecal matter and skin swab samples. They identify Enterococcus faecalis, Staphylococcus epidermidis, and Staphylococcus homini as electrogenic. While the first two bacteria have been previously identified as electrogenic, the main novelty of the manuscript is the implication of glycerol fermentation for electricity production. The authors show that the voltage in a microbial fuel cell increases when glycerol is added and decreases when the 5-methyl furfural is added. Interestingly, glycerol addition enhanced resistance of S. epidermidis to UV-B radiation as well.
This is certainly an interesting topic that merits exploration, but there are a few issues that should be addressed:
1) Regarding Figure 1a-b, the text refers to picking dark colonies to identify electrogenic bacteria. However, in Fig. 1a, it looks like they picked the lightest colonies. In Fig 1b, they all look light. This needs to be explained.
2) There are minor typos and language issues in the paper, although it is otherwise well-written.
3) The ferrozine assay is nice in that many different bacteria can be screened. Why did the authors only pick 4 colonies from each sample? Given how few colonies were tested, I wonder why the authors even bothered with human sampling as opposed to using publicly available strains that are known to be electrogenic. Can the authors at least comment on the likelihood that other taxa besides Staphylococcus and Enterococcus are electrogenic?
Author Response
Responses to reviewers
We appreciate the informative and insightful comments written by the reviewers on our manuscript (Microorganisms-876684) entitled, “Skin Bacteria Mediate Glycerol Fermentation to Produce Electricity and Resist UV-B “. Responses to reviewers point-by-point can be found as below. Changes have been underlined in the text of the manuscript.
Reviewer 1
Comment 1:
Comments and Suggestions for Authors
I would like to congratulate the authors to great and sound study. I enjoyed reading it and consider EET a very important scientific topic. Before publishing I would suggest a few minor changes.
Section 3.3. OD 562nm and Phenol red
Maybe clarify here a bit that the absorbance of phenol red at 562nm is representing the pH of the solution. It can be confusing for readers with different backgrounds to understand this section. For example, a scientist who is used to work with mammalian cells will miss in the Figure 3 the strong red color from phenol red as it is seen in tissue cultures. A microbiologist unfamiliar with phenol red could mix up the OD 562nm with the commonly used OD 600nm which describes turbidity. The microbiologist might then wonder why a decrease in OD562 is representing fermentation. One way could be to speak about absorbance at 562nm instead of OD and then briefly in one sentence clarify that this absorbance represents change in pH before stating the results.
Response 1:
Answer: Sentence has been fixed and highlighted with an additional reference cited.
Line 189,190 :
A change in color of phenol red from red to yellow and a significant reduction of the optical density of 562nm (OD562) due to low pH values [23] in the culture media of the S. epidermidis in the presence of 2% glycerol were detected, indicating the occurrence of bacterial fermentation (Figure 3a).
Reference 23: Kao, M.S.; Huang, S.; Chang, W.L.; Hsieh, M.F.; Huang, C.J.; Gallo, R.L.; Huang, C.M. Microbiome precision editing: Using PEG as a selective fermentation initiator against methicillin-resistant Staphylococcus aureus. Biotechnol J 2017, 12, doi:10.1002/biot.201600399.
Comment 2:
Line 203 it seems the word “glycerol” is missing after the 2%
Response 2:
Answer: The missing word of “glycerol” has been added in the line 203.
“absence or presence of 2% glycerol for 24 h before application onto anode for detection of voltage.”
Comment 3:
In the discussion section the authors suggest "literature has recognized electrons as antioxidants to neutralize free radicals [41]." I find the cited publication weak. It is only a review in "THE JOURNAL OF ALTERNATIVE AND COMPLEMENTARY MEDICINE" which I am unfamiliar with. Also, the main concept of grounding discussed in the review is puzzling to me and I found quite a bit of discussion on this topic. Primary literature on this was never published well. I would therefore suggest to the authors to either provide more citations on the point that free electrons can act as antioxidant, mark it more as a speculation or remove the statement.
In my eyes EET is by itself already an extremely exciting phenomena which reaches in my opinion beyond the simple use as fuel cells. It would not even be necessary to connect this to the antioxidant concept. That this could now be proven in skin commensal bacteria is very exciting and I am looking forward to the follow up work of the authors.
Response 3:
Answer: We totally agree reviewer’s comment. The sentence at the end of Discussion has been rewritten as below.
“The function of electricity produced by S. epidermidis in the human skin microbiome is not yet defined. Although the concept of electrons as antioxidants was proposed [43], further validation of free radical neutralization by electrons produced by bacteria is required.”
Reviewer 2
Comment 1:
Comments and Suggestions for Authors
The authors use a ferrozine assay to identify electrogenic bacteria from human fecal matter and skin swab samples. They identify Enterococcus faecalis, Staphylococcus epidermidis, and Staphylococcus hominis as electrogenic. While the first two bacteria have been previously identified as electrogenic, the main novelty of the manuscript is the implication of glycerol fermentation for electricity production. The authors show that the voltage in a microbial fuel cell increases when glycerol is added and decreases when the 5-methyl furfural is added. Interestingly, glycerol addition enhanced resistance of S. epidermidis to UV-B radiation as well.
This is certainly an interesting topic that merits exploration, but there are a few issues that should be addressed:
1) Regarding Figure 1a-b, the text refers to picking dark colonies to identify electrogenic bacteria. However, in Fig. 1a, it looks like they picked the lightest colonies. In Fig 1b, they all look light. This needs to be explained.
Response 1:
Answer: Identification of Enterococcus faecalis which has known as an electrogenic bacterium validates the feasibility of using TSB agar-based ferrozine overlays for screening the novel electrogenic bacteria in the human microbiome.
We randomly selected four colonies with dark purple complexes and one colony with a light purple complex as a control. Although bacteria formed light purple complexes may be poorly electrogenic in the presence of glucose or glycerol, we cannot rule out those bacteria can mediate other carbon sources for electricity production.
The description has been added onto the end of the first paragraph of Discussion Section.
“Although E. avium in the fecal sample and P. vagans in the skin swab sample formed light purple complexes in the presence of glucose or glycerol (Figure 1), we cannot rule out the possibility that those bacteria can mediate other carbon sources for electricity production.”
Comment 2:
2) There are minor typos and language issues in the paper, although it is otherwise well-written.
Response 2:
Answer: We have fixed minor typos and language issues in revised manuscript.
Comment 3:
3) The ferrozine assay is nice in that many different bacteria can be screened. Why did the authors only pick 4 colonies from each sample? Given how few colonies were tested, I wonder why the authors even bothered with human sampling as opposed to using publicly available strains that are known to be electrogenic. Can the authors at least comment on the likelihood that other taxa besides Staphylococcus and Enterococcus are electrogenic?
Response 3:
Answer: The valuable comment has been added onto the end of first paragraph of Discussion Section. One new reference has been cited.
“Several bacteria such as Bacillus subtilis, Flavobacterium sp., Aeromonas hydrophila, Citrobacter freundii, and Stenotrophomonas sp., have been reported as electrogenic microorganisms [36]. However, most bacteria identified from the TSB agar-based ferrozine overlay assay were Staphylococcus and Enterococcus species. One of possible reasons was that some bacteria with sensitivities to irons may be unable to grow on agar plates which contained ferric ammonium citrate.
Editor
References 35 and 36 are the same.
Please revise, or please make a note to us to edit. Kindly note that we can help to remove the duplicated one and renumber all the references.
Response 1:
Answer: The duplicated references have been removed.
